



# Regional modeling of the Shirase Drainage Basin, East Antarctica: Full-Stokes vs. shallow-ice dynamics

Hakime Seddik[1], Ralf Greve[1], Thomas Zwinger[2], and Shin Sugiyama[1]

[1]Institute of Low Temperature Science, Hokkaido University, Kita-19, Nishi-8, Kita-ku, Sapporo 060-0819, Japan
[2]CSC – IT Center for Science, P.O. Box 405, FIN-02101 Espoo, Finland

*Correspondence to:* H. Seddik
(hakime@lowtem.hokudai.ac.jp)

**Abstract.** A hierarchy of approximations of the force balance for the flow of grounded ice exists, ranging from the most sophisticated full-Stokes (FS) formulation to the most simplified shallow ice approximation (SIA). Both are implemented in the ice flow model Elmer/Ice, and we compare them by applying the model to the East Antarctic Shirase Drainage Basin. First, we apply the control inverse method to infer the distribution of basal friction with FS. We then compare FS and SIA

by simulating the flow of the drainage basin under present-day conditions and for three scenarios 100 years into the future defined by the SeaRISE (Sea-level Response to Ice Sheet Evolution) project. FS reproduces the observed flow pattern of the drainage basin well, in particular the zone of fast flow near the grounding line, while SIA generally overpredicts the surface velocities. As for the transient scenarios, the ice volume change (relative to the constant-climate control run) of the surface climate experiment is nearly the same for FS and SIA, while for the basal sliding experiment (halved basal friction), the ice

volume change is $\sim 30\,\%$ larger for SIA than for FS. This confirms that, in order to model ice sheet areas containing ice streams and outlet glaciers with high resolution and precision, careful consideration must be given to the choice of a suitable force balance.

## 1 Introduction

The Shirase Drainage Basin in East Antarctica covers an area of $2 \times 10^5\,\mathrm{km}^2$. The basin is characterized by ice flow con-

verging into the Shirase Glacier, one of the fastest flowing glaciers in Antarctica. As the glacier calves nearly 90 % of total ice discharged from the basin, the converging flow regime and the fast-flow feature play a crucial role in the mass budget of this region. The dynamic conditions of the drainage basin were extensively measured, with flow speeds at the grounding line between 2300 and $2600\,\mathrm{m\,a^{-1}}$ (Rignot, 2002; Pattyn and Derauw, 2002; Nakamura et al., 2007; Rignot et al., 2011), and an ice mass discharge of about 10 to $14\,\mathrm{Gt\,a^{-1}}$ (Fujii, 1981; Nakamura et al., 2010, 2016). Pattyn and Naruse (2003) studied the

complex ice flow in the Shirase Glacier catchment. They showed that it originates from the Mizuho Triangulation Chain (MTC; Naruse, 1978), and between there and the grounding zone the ice flow regime is mainly dominated by longitudinal stresses and basal sliding.

Due to the nature of the ice flow, the Shirase Drainage Basin is an interesting location for comparing different formulations of ice dynamics. Such comparisons have been the subject of intense research activity in the ice sheet modeling community for sev-





eral years. Extensive initiatives to make comparisons originated from the European Ice Sheet Modeling Initiative (EISMINT) I and II benchmark experiments (Huybrechts et al., 1996; Payne et al., 2000) and the ISMIP-HOM and MISMIP projects (Pattyn et al., 2008, 2012). These initiatives used different representations to compare the ice dynamics for various experimental appli-cations. Another type of comparison was carried out by Mangeney and Califano (1998) where the SIA representation (Hutter,

1983) was used to model the flow on a plane with ice anisotropy under steady state conditions. The results were compared to a full-Stokes (FS) representation. SIA gave excellent results for isotropic ice with uneven bedrock, and acceptable results for anisotropic ice. Comparisons using three-dimensional topographies have also emerged. By inferring the spatial pattern of basal drag using control methods with three different representations of flow (FS; MacAyeal shelfy stream (MacAyeal, 1989) and Blatter-Pattyn representations (Blatter, 1995; Pattyn, 2003)), Morlighem et al. (2010) showed that the simplified ice flow

models suggest high basal drag near the grounding line while the FS model has almost none. (The overestimation with the simplified models might originate from neglecting the bridging effects in an ice stream region of rapidly varying ice thickness). Different models have also been compared for entire ice sheets. Seddik et al. (2012) applied both the Elmer/Ice model, incorpo-rating FS and SICOPOLIS incorporating SIA to the Greenland ice sheet forced on the basis of experiments from the Sea-level Response to Ice Sheet Evolution (SeaRISE) assessment project. They found that FS was much more sensitive for experiments

in which a dynamical forcing (doubled basal sliding) was applied. Similarly, the SeaRISE working group carried out sensitivity experiments for both the Greenland and Antarctica ice sheets with a large set of models and found a large disparity between them (Bindschadler et al., 2013; Nowicki et al., 2013a, b).

Here, we apply the model Elmer/Ice (Gagliardini et al., 2013) to the Shirase Drainage Basin. Elmer/Ice can be run in either FS or SIA mode. This allows us to compare both force balances in the same model and with the same mesh, in contrast to

similar studies in the past (Seddik et al., 2012; Bindschadler et al., 2013; Nowicki et al., 2013a, b). We consider the results obtained with the FS dynamics to be more physically adequate and use them as a reference to infer the basal friction coefficient with the InSAR-based Antarctica ice velocity map provided by Rignot et al. (2011) and a control inverse method. The optimized distribution of the basal friction coefficient is then used to compute the present-day flow of the Shirase Drainage Basin with both FS and SIA, thus allowing a direct comparison between the two methods. We also carry out three transient experiments

100 years into the future, of which the forcings are taken from the suite of SeaRISE experiments. The surface velocities and volume changes produced by FS for each experiment are compared to those produced by SIA. In addition, we investigate the differences in the slip ratios (the ratio of basal to surface velocity) of the FS and SIA solutions.



## 2 Elmer/Ice model

### 2.1 Dynamic/thermodynamic field equations

#### 2.1.1 Full-Stokes (FS) formulation

Since ice is an (almost) incompressible material, conservation of mass entails that the velocity field is solenoidal. Furthermore,
the acceleration (inertial force) is negligible. The full-Stokes (FS) equations are thus

$$\operatorname{div} \boldsymbol{v} = 0, \tag{1}$$

$$\operatorname{div} \mathbf{T} + \rho \boldsymbol{g} = 0, \tag{2}$$

where $\mathbf{T}$ is the Cauchy stress tensor, $\boldsymbol{g} = -g\boldsymbol{e}_z$ the gravitational acceleration vector pointing downward, $\rho$ the density of ice and
$\boldsymbol{v}$ the velocity vector (e.g., Gagliardini et al., 2013). The Cauchy stress tensor can be expressed as $\mathbf{T} = -p\mathbf{I} + \mathbf{T}^{\mathrm{D}}$, where $\mathbf{T}^{\mathrm{D}}$
is the traceless stress deviator, $p = -\operatorname{tr} \mathbf{T}/3$ is the pressure and $\mathbf{I}$ is the unit tensor. Ice rheology is represented by a non-linear
Norton-Hoff-type flow law (commonly referred to as Glen's flow law),

$$\mathbf{T}^{\mathrm{D}} = 2\eta \mathbf{D}, \tag{3}$$

where $\mathbf{D}$ is the strain rate (stretching) tensor. The effective viscosity $\eta$ is defined as

$$\eta = \frac{1}{2} \left[ EA(T') \right]^{-1/n} d^{-(1-1/n)}, \tag{4}$$

where $d$ is the effective strain rate, $A$ is the rate factor and $E$ is the flow enhancement factor (e.g., Gagliardini et al., 2013).

The temperature equation follows from the general balance equation of internal energy and is

$$\rho c(T) \left( \frac{\partial T}{\partial t} + \boldsymbol{v} \cdot \operatorname{grad} T \right) = \operatorname{div} \left( \kappa(T) \operatorname{grad} T \right) + 4\eta d^2 \tag{5}$$

(e.g., Greve and Blatter, 2009), where $\kappa$ and $c$ are the heat conductivity and specific heat of ice, respectively (Table 1).

#### 2.1.2 Shallow ice approximation (SIA)

In contrast to the FS formulation which neglects none of the stress components, SIA assumes that grounded ice flow is governed
only by ice pressure and the vertical shear stresses. This yields the following equations for the velocities $\boldsymbol{v} = (v_x, v_y, v_z)$ and





the pressure $p$ (Greve and Blatter, 2009),

$$\frac{\partial v_x}{\partial z} = -\left(\frac{\rho g}{\eta}\right)^n (h-z)^n \left[\sqrt{\left(\frac{\partial h}{\partial x}\right)^2 + \left(\frac{\partial h}{\partial y}\right)^2}\right]^{n-1} \frac{\partial h}{\partial x},$$

$$\frac{\partial v_y}{\partial z} = -\left(\frac{\rho g}{\eta}\right)^n (h-z)^n \left[\sqrt{\left(\frac{\partial h}{\partial x}\right)^2 + \left(\frac{\partial h}{\partial y}\right)^2}\right]^{n-1} \frac{\partial h}{\partial y},$$

$$\frac{\partial v_z}{\partial z} = -\frac{\partial v_x}{\partial x} - \frac{\partial v_y}{\partial y},$$

$$\frac{\partial p}{\partial z} = -\rho g, \tag{6}$$

where, in the Cartesian coordinate system $(x, y, z)$ with $z$ positive upward, $z = h(x, y, t)$ denotes the free surface. The viscosity term $\eta$ is computed with Eq. (4) and the SIA effective strain rate

$$d = \frac{1}{2}\sqrt{\left(\frac{\partial v_x}{\partial z}\right)^2 + \left(\frac{\partial v_y}{\partial z}\right)^2}. \tag{7}$$

Equations (6) are solved by formulating a degenerated Poisson equation that is discretized with the finite element method (Appendix A).

The SIA formulation of the temperature equation (Eq. (5)) follows from the negligible horizontal heat conduction assumption. Then, by expressing the effective strain rate $d$ in terms of the effective stress $\sigma_e$ in the dissipation term (Greve and Blatter, 2009), the equation is

$$\rho c(T)\left(\frac{\partial T}{\partial t} + \boldsymbol{v} \cdot \operatorname{grad} T\right) = \frac{\partial}{\partial z}\left(\kappa(T)\frac{\partial T}{\partial z}\right) + 2A(T')\sigma_e^{n+1}. \tag{8}$$

## 2.2 Boundary conditions

### 2.2.1 Ice surface

For both FS and SIA, we assumed a stress-free ice surface (atmospheric pressure and wind stress neglected). In the case of FS, the evolution of the upper surface $h(x, y, t)$ is governed by the kinematic boundary condition,

$$\frac{\partial h}{\partial t} + v_x \frac{\partial h}{\partial x} + v_y \frac{\partial h}{\partial y} - v_z = a_s, \tag{9}$$

where $a_s(x, y, t)$ is the accumulation-ablation function or surface mass balance. The free surface evolution equation employed in SIA results from the ice thickness equation (Greve and Blatter, 2009),

$$\frac{\partial h}{\partial t} = \frac{\partial}{\partial x}\left(D\frac{\partial h}{\partial x}\right) + \frac{\partial}{\partial y}\left(D\frac{\partial h}{\partial y}\right) + a_s - a_b + \frac{\partial b}{\partial t}, \tag{10}$$

where $a_b$ is the basal melting rate, $z = b(x, y, t)$ is the ice base, and $D$ is defined by the function

$$D = \frac{\rho g H^2}{\beta} + 2(\rho g)^n |\operatorname{grad} h|^{n-1} \int_b^h EA(T')(h-z)^{n+1}\, dz, \tag{11}$$





where $H$ is the ice thickness and $\beta$ is the basal friction coefficient (Sect. 2.3).

The mean annual surface temperature $T_{\mathrm{ma}}$ and the mean summer surface temperature $T_{\mathrm{ms}}$ for Antarctica were parameterized as functions of surface elevation $h$, latitude $\phi$ and time $t$. Following Fortuin and Oerlemans (1990), the current mean annual surface temperature is

$$
\begin{aligned}
T_{\mathrm{ma,present}}(\phi,h) &= d_{\mathrm{ma1}} + \gamma_{\mathrm{ma1}}h + c_{\mathrm{ma1}}\phi && (h > 1500\,\mathrm{m}), \\
T_{\mathrm{ma,present}}(\phi,h) &= d_{\mathrm{ma2}} + \gamma_{\mathrm{ma2}}h + c_{\mathrm{ma2}}\phi && (h = 200 - 1500\,\mathrm{m}), \\
T_{\mathrm{ma,present}}(\phi,h) &= d_{\mathrm{ma3}} + c_{\mathrm{ma3}}\phi && (h = 0 - 200\,\mathrm{m}),
\end{aligned}
\tag{12}
$$

where the temperature constants are $d_{\mathrm{ma1}} = 7.405°\mathrm{C}$, $d_{\mathrm{ma2}} = 36.689°\mathrm{C}$, $d_{\mathrm{ma3}} = 49.642°\mathrm{C}$, the mean lapse rates are $\gamma_{\mathrm{ma1}} = -0.014285°\mathrm{C\,m}^{-1}$, $\gamma_{\mathrm{ma2}} = -0.005102°\mathrm{C\,m}^{-1}$ and the latitude coefficients are $c_{\mathrm{ma1}} = -0.108°\mathrm{C}\,(°\mathrm{S})^{-1}$, $c_{\mathrm{ma2}} = -0.725°\mathrm{C}\,(°\mathrm{S})^{-1}$, $c_{\mathrm{ma3}} = -0.943°\mathrm{C}\,(°\mathrm{S})^{-1}$. Following Huybrechts and de Wolde (1999), the current mean summer surface temperature is

$$
T_{\mathrm{ms,present}}(\phi,h) = d_{\mathrm{ms}} + \gamma_{\mathrm{ms}}h + c_{\mathrm{ms}}\phi,
\tag{13}
$$

where the temperature constant is $d_{\mathrm{ms}} = 16.81°\mathrm{C}$, the mean lapse rate is $\gamma_{\mathrm{ms}} = -0.00692°\mathrm{C\,m}^{-1}$ and the latitude coefficient is $c_{\mathrm{ms}} = -0.27937°\mathrm{C}\,(°\mathrm{S})^{-1}$. For climatic conditions different from the current, mean annual surface temperature was modified by the additive anomaly $\Delta T_{\mathrm{ma}}(\phi,h,t)$ and mean summer surface temperature by $\Delta T_{\mathrm{ms}}(\phi,h,t)$.

The surface mass balance $a_{\mathrm{s}}$ (snowfall minus melting) was computed with the SeaRISE specifications for Antarctica (Sato and Greve, 2012; Bindschadler et al., 2013; Nowicki et al., 2013a). For the current mean annual precipitation rate, $P_{\mathrm{ma,present}}(\lambda,\phi)$ (function of longitude $\lambda$ and latitude $\phi$), we used the data from Le Brocq et al. (2010) (newly compiled from Arthern et al. (2006)). For climatic conditions different from the current, they were modified by multiplicative anomalies (factors) $f_{P_{\mathrm{ma}}}(\lambda,\phi,t)$. Surface melting was parameterized by a positive degree day (PDD) method (Reeh, 1991) supplemented by the semi-analytical solution for the PDD integral (Calov and Greve, 2005). The PDD factors were $\beta_{\mathrm{ice}} = 8\,\mathrm{mm}$ ice equiv. $\mathrm{d}^{-1}\,°\mathrm{C}^{-1}$ for ice melt and $\beta_{\mathrm{snow}} = 3\,\mathrm{mm}$ ice equiv. $\mathrm{d}^{-1}\,°\mathrm{C}^{-1}$ for snow melt (Huybrechts and de Wolde, 1999). Furthermore, the standard deviation of short-term, statistical air temperature fluctuations was $\sigma = 5°\mathrm{C}$ (Huybrechts and de Wolde, 1999), and the saturation factor for the formation of superimposed ice was chosen as $P_{\mathrm{max}} = 0.6$ (Reeh, 1991). Conversion from precipitation to snowfall (solid precipitation) was done on a monthly basis with the empirical relation by Marsiat (1994).

### 2.2.2 Ice bed

On the short time-scales of our simulations, the bed topography $b(x,y)$ can be assumed to be rigid (isostatic compensation neglected) and thus at all times equal to the prescribed initial condition (the last term in the right-hand side of Eq. (10) therefore vanishes). For FS, basal shear stresses and basal velocities are related according to

$$
\boldsymbol{t}^{\mathrm{T}} \cdot \mathbf{T} \cdot \boldsymbol{n} + \beta \boldsymbol{v} \cdot \boldsymbol{t} = 0,
\tag{14}
$$





where $\beta$ is the basal friction coefficient (Sect. 2.3), $\boldsymbol{n}$ is the normal unit vector and $\boldsymbol{t}$ is the tangential unit vector aligned with the flow direction. For the SIA, Eq. (14) simplifies to a statement about the horizontal components of the basal velocity,

$$v_x = -\frac{\rho g H}{\beta} \frac{\partial h}{\partial x},$$
$$v_y = -\frac{\rho g H}{\beta} \frac{\partial h}{\partial y}. \tag{15}$$

In both cases, the normal velocity at the ice–bed interface is given by

$$\boldsymbol{v} \cdot \boldsymbol{n} = a_{\mathrm{b}}, \tag{16}$$

where the basal meting rate $a_{\mathrm{b}}$ is determined by the energy jump condition (Greve and Blatter, 2009)

$$a_{\mathrm{b}} = \frac{q_{\mathrm{geo}} - \kappa(T)\,(\mathrm{grad}\,T \cdot \boldsymbol{n}) - \boldsymbol{v} \cdot \sigma \cdot \boldsymbol{n}}{\rho L}, \tag{17}$$

where $L$ is the latent heat of ice and $q_{\mathrm{geo}}$ the spatially varying geothermal flux (Shapiro and Ritzwoller, 2004).

Finally, the temperature equations (Eqs. (5), (8)) were solved assuming a temperate base everywhere (see below, Sect. 2.4). This yields the Dirichlet-type condition

$$T = T_{\mathrm{m}}, \tag{18}$$

where $T_{\mathrm{m}}$ is the temperature at the pressure melting point. The pressure $p$ for SIA was assumed to be hydrostatic ($p = \rho g(h-z)$) in the calculation of $T_{\mathrm{m}}$.

### 2.2.3 Side boundaries

Our computational domain (see below, Sect. 2.4, for details) has two different types of side boundaries, namely two lateral flow lines and a downstream grounding line. For the FS case, we assumed vanishing normal velocities and zero stress conditions for the flow lines,

$$\boldsymbol{v} \cdot \boldsymbol{n} = 0,$$
$$\boldsymbol{n}^{\mathrm{T}} \cdot \mathbf{T} \cdot \boldsymbol{n} = 0, \tag{19}$$
$$\boldsymbol{t}^{\mathrm{T}} \cdot \mathbf{T} \cdot \boldsymbol{n} = 0.$$

The additional condition

$$\boldsymbol{v} \cdot \boldsymbol{t} = 0 \quad (H < 10\,\mathrm{m}) \tag{20}$$

(zero tangential velocity) was only applied to nodes with an ice thickness of less than $10\,\mathrm{m}$, otherwise a tangential free slip condition was assumed. At the grounding line, the normal component of the stress vector is equal to the hydrostatic pressure exerted by the ocean for nodes under the water line and zero elsewhere, and the tangential component is zero:

$$\boldsymbol{n}^{\mathrm{T}} \cdot \mathbf{T} \cdot \boldsymbol{n} = -\max(\rho_{\mathrm{w}} g(l_{\mathrm{w}} - z), 0),$$
$$\boldsymbol{t}^{\mathrm{T}} \cdot \mathbf{T} \cdot \boldsymbol{n} = 0, \tag{21}$$





where $\rho_{\mathrm{w}} = 1025\,\mathrm{kg\,m^{-3}}$ is the sea water density and $l_{\mathrm{w}}$ is the sea level.

For the SIA case, neither of the dynamic conditions (19)–(21) are required. Instead, a boundary condition for the free surface evolution equation (10) is needed. For the lateral flow lines, we assume a zero surface gradient in normal direction,

$$\mathrm{grad}\,h \cdot \boldsymbol{n} = 0, \tag{22}$$

which, in the SIA, is equivalent to the zero-normal-velocity condition (Eq. (19)$_1$). At the grounding line, we keep the ice surface $h$ fixed at its observed, present-day distribution.

## 2.3 Control inverse method

The control inverse method, introduced by MacAyeal (1993), was implemented for the FS mode of Elmer/Ice by Gillet-Chaulet et al. (2012). It relies on the computation of the FS adjoint (Morlighem et al., 2010) and on the definition of a cost function

expressed as the difference between the norm of the modeled and observed horizontal velocities ($\boldsymbol{v}_{\mathrm{h}}$ and $\boldsymbol{v}_{\mathrm{h}}^{\mathrm{obs}}$, respectively):

$$J_{\mathrm{o}} = \int_{\Gamma_s} \frac{1}{2}(|\boldsymbol{v}_{\mathrm{h}}| - |\boldsymbol{v}_{\mathrm{h}}^{\mathrm{obs}}|)^2\,\mathrm{d}\Gamma. \tag{23}$$

The regularization procedure is constructed by adding a smoothness constraint to the cost function in the form of a Tikhonov regularization term,

$$J_{\mathrm{reg}} = \frac{1}{2}\int_{\Gamma_b} \left(\frac{\partial \alpha}{\partial x}\right)^2 + \left(\frac{\partial \alpha}{\partial y}\right)^2\,\mathrm{d}\Gamma, \tag{24}$$

where the parameter $\alpha$ is used to avoid negative values of the basal friction coefficient $\beta$ and is defined as

$$\beta = 10^{\alpha}. \tag{25}$$

The total cost function is then given by

$$J_{\mathrm{tot}} = J_{\mathrm{o}} + \lambda J_{\mathrm{reg}}, \tag{26}$$

where $\lambda$ is a positive ad-hoc parameter. Minimizing the cost function $J_{\mathrm{tot}}$ with respect to $\alpha$ was implemented using the quasi-

Newton routine M1QN3 (Gilbert and Lemaréchal, 1989; Gillet-Chaulet et al., 2012).

## 2.4 Computational domain and meshing

The Elmer/Ice model was applied to the Shirase Drainage Basin. The present-day surface and bed topographies were extracted from the Bedmap2 data set by Fretwell et al. (2013). Our domain originates at Dome Fuji, is delimited by two approximate flow lines and converges into Shirase Glacier (Fig. 1). The downstream end of the domain is the grounding line of Shirase

Glacier, which means that we have removed the small ice shelf of the glacier flowing into Havsbotn (the bay that forms the head of Lützow-Holm Bay). The bed topography (Fig. 2) shows by trend a decrease in elevation from the interior of the ice sheet towards the coast, and the bed of the most downstream regions including Shirase Glacier is below sea level.



For this domain, we solved Equations (1)-(5) for the FS case and Equations (6)-(8) for the SIA case with the finite element method. Since radar data indicate that the basal temperature in the area is at the melting point everywhere except for the highest peaks of the bed topography (Fujita et al., 2012), for simplicity we assumed a temperate base for the entire domain (Eq. (18)). The finite element mesh was constructed by using an adaptive meshing method with horizontal resolutions up to 200 m near the grounding line, 300 m near Dome Fuji (Fig. 3) and coarser resolutions down to 15 km elsewhere. We first constructed a 2D footprint mesh and then extended it to give a 3D mesh with 56394 elements and 11 equidistant, terrain-following layers.

The non-linearity of the model equations was dealt with by Picard iteration as in Seddik et al. (2012). Stabilization methods (Franca et al., 1992; Franca and Frey, 1992) were applied to the finite element discretization. The resulting system of linear equations was solved with a direct method using the parallel sparse direct solver MUMPS (Amestoy et al., 2001, 2006). The position of the ice front was assumed to be fixed. A minimum ice thickness of 10 m was applied everywhere and for all times. This means that locations that are initially glaciated were not allowed to become completely ice-free but rather are always covered with at least 10 m of ice.

# 3 Numerical experiments

## 3.1 Present-day state

In order to infer an initial spatial distribution of the temperature field that contains an historical footprint of ice sheet evolution during glacial cycles, a spin-up of the whole ice sheet is generally needed. However, this procedure does not always produce a distribution in satisfactory agreement with present conditions, particularly at the ice base (Sato and Greve, 2012), and the simulation periods of a glacial cycle needed for a proper spin-up state implies the need to use an SIA model to reduce the computational cost. Instead, here we used a simple 1D steady state approach to initialize the temperature distribution. Equation (5) simplifies to

$$\rho c(T) v_z \frac{\partial T}{\partial z} = \frac{\partial}{\partial z}\left(\kappa(T)\frac{\partial T}{\partial z}\right). \tag{27}$$

Following Greve et al. (2002), the vertical velocity $v_z$ in Eq. (27) was computed assuming a Dansgaard-Johnsen distribution (Dansgaard and Johnsen, 1969). This consists of a constant vertical strain rate $\partial v_z/\partial z$ from the free surface down to $z = 0.25H$, and a linearly decreasing vertical rate below.

Using the temperature field computed by solving Eq. (27) for each ice column of the domain, we applied the control inverse method (Sect. 2.3) to infer in FS the spatial distribution of the basal friction coefficient $\beta$ (starting with a spatially uniform value of $\beta = 10^{-4}\,\mathrm{MPa\,m^{-1}\,a}$ as initial guess) and the present-day 3D velocity field. The observed velocities used for the inversion were given by the InSAR-based Antarctica ice velocity map (Rignot et al. (2011) and Fig. 3), and the basal melting rate $a_\mathrm{b}$ in the boundary condition (16) was neglected (thus the normal velocity at the ice base was set to zero). For comparison, we also computed the SIA velocity field based on the same distribution of $\beta$.





## 3.2 Future climate experiments

The obtained present-day state of the Shirase Drainage Basin served as initial condition for runs into the future. We used a subset of the SeaRISE experiments defined in Bindschadler et al. (2013) and Nowicki et al. (2013a):

- Experiment CTL ("constant-climate control run"):

  Starting at the present (more precisely, the epoch 2004-1-1 0:00 corresponding to $t = 0$) and running for 100 years, with the climate held at the current state.

- Experiment S1 ("basal sliding experiment"):

  Constant climate forcing with increased basal lubrication. This was implemented in Elmer/Ice (for both FS and SIA) by halving the basal friction coefficient (approximately doubling the basal sliding) everywhere in the domain.

- Experiment C2 ("surface climate experiment"):

  $1.5 \times$ AR4 surface climate forcing; starting with the current state as in CTL and S1, but with climatic forcing (changes in mean annual temperature, mean summer temperature and precipitation; see Sect. 2.2.1). The changes were derived from an ensemble average of 18 AR4 models, run for the period 2004-98 under the A1B emission scenario. For runs beyond 2098, conditions were held at those of 2098.

Note that, as defined by SeaRISE, the combination experiment R8 designed for the IPCC AR5 does not include increased basal sliding during the first 100 years and the increase in basal melting is applied only to the Amundsen Sea and the Amery Ice Shelf. Therefore, experiment C2 is equivalent to R8 for the Shirase Drainage Basin and the period considered here. All runs were carried out in both FS and SIA mode. For the FS runs, the numerical time step was $\Delta t = 0.02\,\mathrm{a}$ for the first 5 years model time and $\Delta t = 0.05\,\mathrm{a}$ thereafter, while for the SIA runs it was $\Delta t = 0.05\,\mathrm{a}$ for the entire 100 years.

## 4 Results

### 4.1 Present-day state

As described above (Sect. 3.1), we used the control inverse method to compute the spatial distribution of the basal friction coefficient and the 3D velocity field in FS. The parameter $\lambda$ in Eq. (26) was chosen using the L-curve method (Hansen, 2001; Gillet-Chaulet et al., 2012) which plots the regularization, i.e., the term $J_{\mathrm{reg}}$ as a function of the term $J_{\mathrm{o}}$ in Eq. (26). The L-curve obtained with the control inverse method is shown in Fig. 4. When we increased $\lambda$ from 0 to $10^{10}$, the roughness of the basal friction coefficient distribution, represented by $J_{\mathrm{reg}}$, decreased by several orders of magnitude, while the mismatch between the observed and the computed velocities ($J_{\mathrm{o}}$) remained essentially constant. For higher values of $\lambda$, the roughness $J_{\mathrm{reg}}$ continued to decrease; however, at the cost of an increasing $J_{\mathrm{o}}$ as the basal friction coefficient distribution became too smooth. We therefore chose $\lambda = 10^{10}$ as the optimal value.

Figures 5a–c show the comparison between the observed velocities and the velocities computed using FS and the control inverse method. The overall agreement is good; in particular, the model reproduced well the fast-flowing ice towards the





grounding line and the slower ice motion elsewhere in the drainage basin. The most significant mismatch occurs at the upstream end of the domain in the vicinity of Dome Fuji where ice flow is slowest. This is due to the formulation of the control inverse method. The cost function works better in regions with high velocities than in slow-moving areas because of the quadratic dependence of Eq. (23) on the velocity mismatch. This leads to a larger contribution of mismatches that occur at high velocities

to the cost function, while the contribution of mismatches that occur at low velocities is limited. Another notable mismatch occurs in the right (eastern) part of the domain about one half to two thirds in downstream direction.

The spatial pattern of the basal friction coefficient obtained by the optimization (Fig. 5d) is characterized by a region with low friction at Shirase Glacier and the adjacent, downstream area of the drainage basin. Further upstream, the basal friction generally increases, with the exception of the vicinity of Dome Fuji where the friction is again low. As stated above, the latter

result is not very relevant because the cost function is rather insensitive to velocity mismatches in this slowest flowing area, which limits the quality of the obtained friction coefficients. Along the lateral boundaries of the domain, the basal friction is also low. This is likely related to the choice of the boundary conditions (Eq. (19)) and thus an artefact rather than a real phenomenon.

An initial velocity distribution was also computed with SIA, using the basal friction coefficient inferred from FS and the

control inverse method (Fig. 5d). Figure 6a shows the resulting surface velocities. They were generally higher than those for FS (see also Fig. 6b). In particular, SIA did not reproduce so well the sharp contrast between the narrow fast-flowing region near the grounding line and the slower flowing ice further upstream. Figures 6c,d show the surface velocity ratio FS/SIA vs. the FS slip ratio (panel c) and the SIA slip ratio (panel d). In both cases, the scatter of the surface velocity ratio increases as the slip ratio increases from near zero to unity. This is because the SIA is best suited for areas where the slip ratio is low and the

flow regime is dominated by vertical shear, while in areas with a high slip ratio near-plug-flow conditions prevail for which the SIA is less appropriate (e.g., Blatter et al., 2011).

## 4.2 Future climate experiments

These experiments with evolving ice surface were carried out with the previously computed present-day state as initial condition. For both FS and SIA, the distribution of the basal friction coefficient obtained by FS and the control inverse method was

used (Fig. 5d), and it was kept constant with time.

Figure 7 depicts the simulated evolution of the ice volume ($V$) of the Shirase Drainage Basin (panel a) and the volume relative to CTL (panel b) for the three scenarios. For CTL-FS (Fig. 7a, solid blue line), the average ice volume change (net mass balance) in the first year of the simulation is $+2.0 \times 10^{-3}\,\mathrm{mm\,SLE\,a^{-1}}$, and in the first five years it is $-14.6 \times 10^{-3}\,\mathrm{mm\,SLE\,a^{-1}}$. So the mass balance is very slightly positive initially, but then turns to a negative value. Both values are consistent with the

observed mass balance reported by Nakamura et al. (2016), $-1.9 \pm 3.3\,\mathrm{Gt\,a^{-1}}$, which is equivalent to $-5.3 \times 10^{-3} \pm 9.1 \times 10^{-3}\,\mathrm{mm\,SLE\,a^{-1}}$. By contrast, the initial ice volume change produced by CTL-SIA (Fig. 7a, dashed blue line) is much more negative than that of CTL-FS, and by an order of magnitude larger than the most negative value in the range of uncertainty given by Nakamura et al. (2016).





For both FS and SIA, all scenarios produce a volume loss over the 100 years model time. However, it is much smaller for FS than for SIA, so that the results are more dependent on the used model dynamics than on the scenario. The more rapid volume loss in the SIA experiments is certainly related to the larger flow velocities produced by the SIA for the initial state (see above).

The difference between FS and SIA becomes much smaller if we consider volume changes relative to the respective control run ($\Delta V$) rather than absolute volumes (Fig. 7b). It is evident that the basal sliding experiment S1 produces a much larger response than the surface climate experiment C2 over the entire model time. This finding agrees qualitatively with the simulated response of the entire Antarctic ice sheet to these forcings (Bindschadler et al., 2013). For both FS and SIA, C2 produces first a slightly lower volume than CTL, but $\Delta V$ goes through a minimum after $\sim 50\,\mathrm{a}$ and catches up thereafter. Apparently, after $\sim 50\,\mathrm{a}$, the increased precipitation of the surface climate scenario C2 outweighs the initially dominant impact of warming and surface melting. After $100\,\mathrm{a}$, the volume changes are

- S1 (basal sliding exp.) – CTL (control):
  $\Delta V_{\mathrm{FS}} \sim -3.2$ mm SLE, $\Delta V_{\mathrm{SIA}} \sim -4.4$ mm SLE.

- C2 (surface climate exp.) – CTL (control):
  $\Delta V_{\mathrm{FS}} \sim -0.06$ mm SLE, $\Delta V_{\mathrm{SIA}} \sim +0.04$ mm SLE

(where SLE means sea level equivalents). Thus, S1-SIA shows a $\sim 30\,\%$ larger volume change[1] than S1-FS, while the difference is insignificant for C2-SIA vs. C2-FS.

Figure 8 shows the surface velocities and slip ratios (ratio of basal to surface velocity) computed with FS dynamics for each scenario at $t = 100$ years. The distribution of the surface velocities obtained with the control experiment (CTL; Fig. 8a) is similar to the initial velocities computed with the control inverse method (Fig. 5b). The main difference occurs near Dome Fuji where the ice flow has slowed down, which agrees better with the expected slow flow in the vicinity of topographic domes. The slip ratio (Fig. 8b) is characterized by a complex fine structure, but increases generally from the interior ice sheet towards the grounding line. In the narrow channel of Shirase Glacier, the slip ratio is close to unity, which means that the fast flow of the glacier is controlled mainly by basal sliding (as expected). Experiment S1 (halved basal friction) produces a marked acceleration of the ice flow (Fig. 8c), particularly near the grounding line. The slip ratio distribution (Fig. 8d) indicates that basal sliding contributes significantly to the ice flow for practically the entire drainage basin. By contrast, experiment C2 does not differ greatly from CTL (the surface velocities and slip ratios are similarly distributed; Fig. 8e and f). The slip ratios of all three experiments exhibit boundary effects along the lateral margins of the domain. These are artefacts resulting from the low basal friction near the margins (see above, Sect. 4.1).

The surface velocities and slip ratios obtained with SIA dynamics are shown in Fig. 9. The differences compared to the respective FS solutions are notable. For all three scenarios, SIA produces higher surface velocities than FS. Further, the sharp contrast between the narrow fast-flowing region of Shirase Glacier and the slower flowing ice further upstream is not so well reproduced; the zone of fast flow is smeared out over a larger area. This is the same behaviour we found for the control inverse method (Fig. 6a). The slip ratio distributions of the SIA solutions are also very different from those of FS. Not considering

---

[1]Computed as $(\Delta V_{\mathrm{FS}} - \Delta V_{\mathrm{SIA}})/[\frac{1}{2}(\Delta V_{\mathrm{FS}} + \Delta V_{\mathrm{SIA}})]$

the vicinity of Dome Fuji (where the distribution of the basal friction coefficient $\beta$ obtained by the control inverse method is not well constrained, see Sect. 4.1), the slip ratios show generally lower values. The only exception is the channel of Shirase Glacier where the slip ratios approach unity, similar to the FS solutions. However, as we have seen above, the SIA solution is more sensitive to the S1 perturbation (halved basal friction) than the FS solution. Therefore, the smaller SIA slip ratios are

not due to less basal sliding, but rather due to more internal deformation. The latter results from the fact that, in the SIA, the flow is controlled by the local surface topography only and does not experience any resistance due to longitudinal stresses and horizontal shear, which is accounted for by the FS dynamics.

## 5   Discussion

Our finding that the volume change of the Shirase Drainage Basin relative to CTL is small for the surface climate experiment

C2 is consistent with the SeaRISE findings. For both the entire Antarctic ice sheet (Bindschadler et al., 2013) and the QMD (extended Queen Maud Land) basin that includes the Shirase Drainage Basin (Nowicki et al., 2013a), the SeaRISE results for six different models showed a generally small sensitivity of the simulated ice volume to this forcing with an even unclear sign (some models predicted a volume increase, others a decrease). By contrast, all SeaRISE models produced a significant volume decrease for the S1 basal sliding experiment, which also agrees with our results. For the entire Antarctic ice sheet (AIS), the

range of SeaRISE S1 results after 100 model years was $\sim -300$ to $-75\,\mathrm{mm\,SLE}$ (Bindschadler et al., 2013). Scaling this down to the Shirase Drainage Basin (SDB) by simply using the volume ratio $V_{\mathrm{SDB}}/V_{\mathrm{AIS}} \approx 1.148\,\mathrm{m\,SLE}/58.3\,\mathrm{m\,SLE} \approx 0.02$ ($V_{\mathrm{SDB}}$: our value, see Fig. 7a; $V_{\mathrm{AIS}}$ by Vaughan et al. (2013)) yields a range of $\sim -6$ to $-1.5\,\mathrm{mm\,SLE}$. The volume changes of S1 relative to CTL after 100 years we show in Fig. 7b are well in the centre of this range for both FS and SIA.

Especially in the fast-flowing region near the grounding line, the Shirase Drainage Basin is characterized by a complex

stress regime (Naruse, 1978; Pattyn and Naruse, 2003) that the SIA force balance cannot capture fully owing to its neglect of longitudinal stresses and horizontal shear. We investigate the difference between FS and SIA further by considering the stress ratio

$$\varsigma = \frac{\sqrt{\tau_{xz}^2 + \tau_{yz}^2}}{\sigma_{\mathrm{e}}}, \tag{28}$$

where the effective stress $\sigma_{\mathrm{e}}$ is

$$\sigma_{\mathrm{e}} = \sqrt{\tau_{xz}^2 + \tau_{yz}^2 + \tau_{xy}^2 + (\tau_{xx}^{\mathrm{D}})^2 + (\tau_{yy}^{\mathrm{D}})^2 + \tau_{xx}^{\mathrm{D}}\tau_{yy}^{\mathrm{D}}}, \tag{29}$$

and the $\tau_{ij}$ and $\tau_{ij}^{\mathrm{D}}$ are the components of the stress tensor $\mathbf{T}$ and deviator $\mathbf{T}^{\mathrm{D}}$, respectively (see Sect. 2.1.1). Since the longitudinal stresses $\tau_{xx}^{\mathrm{D}}$ and $\tau_{yy}^{\mathrm{D}}$ as well as the horizontal shear stress $\tau_{xy}$ are neglected in the SIA, the SIA stress ratio is equal to unity everywhere. Thus, the deviation of the FS stress ratio from unity reveals where the stress components neglected in the SIA method play an important role for the dynamics of the drainage basin.

Figure 10 depicts the stress ratio at the base of the Shirase Drainage Basin that results from the FS solution for the control run CTL (stress ratios for S1 and C2 are not shown because they are very similar). The stress ratio is close to unity everywhere





except for two regions, namely in the vicinity (several 10s of km) of Dome Fuji and in the narrow trough near the grounding line where Shirase Glacier is located. This numerical finding agrees perfectly with the theoretical understanding that the ice flow regime deviates most from simple, bed-parallel shear (and thus the SIA is violated) in the vicinity of ice domes and for fast-flowing ice streams with high slip ratios (e.g., Blatter et al., 2011).

As described above (Sects. 4.1 and 4.2), for both the FS and SIA experiments we used the distribution of the basal friction coefficient obtained by inverting the FS problem (Fig. 5d). While this approach facilitates comparison between the FS and SIA results, it clearly favours FS. It is therefore not surprising that the constant-climate control run CTL produces a much smaller volume change in FS than in SIA (Fig. 7a). In order to investigate this issue further, we also attempted to invert the SIA problem for basal friction. Owing to the prescribed ice geometry, temperature field and the local nature of the SIA flow field (Eq. (6)),

this is a straightforward exercise that does not require the control inverse method or anything similarly sophisticated. However, the inversion failed to produce meaningful results. For most parts of the domain, even for no-slip conditions at the base the SIA produces surface velocities that exceed the observed ones (Fig. 5a), so that a minimization of the misfit between modeled and observed surface velocities by choosing an optimal distribution of the basal friction coefficient could not be achieved.

We have seen in Sect. 4.2 that, when looking at the ice volume changes relative to the control run CTL, the results of the

scenarios C2 and S1 are not overly different when FS and SIA are compared. For the surface climate scenario C2, they are almost identical, and for the basal sliding scenario S1, SIA shows a $\sim 30\,\%$ larger volume change than FS. However, the caveat of this statement is that we applied different boundary conditions for FS and SIA at the grounding line, which result from the different mathematical nature of the respective systems of partial differential equations (Sect. 2.2.3) and likely influence the solutions to some extent. Further, we compared only the flow dynamics for grounded ice and assumed a fixed grounding line

for Shirase Glacier. Therefore, we did not account for the potentially important impacts of grounding line migration and ice shelf buttressing on the dynamics of the system. Such effects are beyond the scope of the SIA and require at least some flavour of higher-order dynamics.

## 6   Conclusion

We compared two approaches to represent ice flow dynamics for the Shirase Drainage Basin, namely the full-Stokes (FS)

formulation and the shallow ice approximation (SIA), implemented within the same dynamic/thermodynamic ice flow model (Elmer/Ice). The complex nature of the stress regime in the drainage basin allows a good characterization of the differences in the evolution and dynamics of the area resulting from the two approaches. In the first step, we applied an inverse method to infer the distribution of the basal friction coefficient with FS. We then compared FS and SIA by assessing the respective response of the drainage basin to different climatic and dynamic and forcings.

There were evident differences in the computed surface velocities between the two approaches. The surface velocities computed with FS showed a distinct, well-defined fast-flowing area near the grounding line. A similar flow feature is observed in current surface velocities, essentially coinciding with Shirase Glacier. In contrast, the SIA produced a less well defined contrast between the narrow fast-flowing region of Shirase Glacier and the surrounding, slower flowing ice; the zone of fast flow





is distributed over a larger area. In general, the SIA overpredicted the surface velocities everywhere in the domain, which is a consequence of the neglected longitudinal stresses and horizontal shear stress that can generate an efficient resistance to ice flow. Consequently, in transient scenarios, SIA runs consistently produced smaller ice volumes than FS runs. However, when considering ice volume evolution relative to a control run, the difference between the FS and SIA results was not overly large.

Nevertheless, our findings show clearly that FS is superior to the SIA in modeling the ice flow in the area, in particular in fast-flowing regions with high slip ratios.

In this study, we considered grounded ice only and kept the grounding line fixed. A desirable extension would be to include floating ice (the small ice shelf attached to Shirase Glacier) and compare FS to SIA/SSA (SSA: Shallow Shelf Approximation) dynamics within the same model. This would reveal whether the complex interactions between grounded and floating ice,

including grounding line dynamics, lead to further differences in the response of the system to external forcings.

In any case, our findings have implications beyond the immediate application to the Shirase Drainage Basin. It can be expected that FS and SIA will also produce different results for other areas with complex stress regimes and bed topographies. We therefore conclude that careful consideration must be given to the representation of ice flow physics when attempting to model the dynamics and evolution of ice sheet areas containing ice streams and outlet glaciers with high resolution and

precision.

## Appendix A: Solutions of the SIA equations with the finite element method

The SIA is implemented into the finite element method by solving the degenerated Poisson equation

$$\frac{\partial^2 U}{\partial z^2} = \Psi(x,y,z),$$ (A1)

where the field $U(x,y,z)$ is sought and $\Psi(x,y,z)$ is given. This equation is subject to the boundary conditions

$$\begin{aligned} \frac{\partial U}{\partial z} &= \Gamma(x,y), &\text{for } z = \Omega_{\mathrm{f}}(x,y), \\ U &= \bar{U}(x,y), &\text{for } z = \Omega_{\mathrm{u}}(x,y), \end{aligned}$$ (A2)

where $\Gamma(x,y)$ and $\bar{U}(x,y)$ are given functions, and $z = \Omega_{\mathrm{f}}(x,y)$ and $z = \Omega_{\mathrm{u}}(x,y)$ are given boundary surfaces. For the velocity components $v_x$, $v_y$, $v_z$ and the pressure $p$ (Eq. (6)), the following identifications hold:

$$\underline{U = v_x}: \quad \Psi = n\left(\frac{\rho g}{\eta}\right)^n (h-z)^{n-1} \left[\sqrt{\left(\frac{\partial h}{\partial x}\right)^2 + \left(\frac{\partial h}{\partial y}\right)^2}\right]^{n-1} \frac{\partial h}{\partial x}\left(1 + \frac{\partial \eta}{\partial z}\frac{h-z}{\eta}\right),$$ (A3)

with $\Gamma = 0$ on $z = h$, and $\bar{U} = v_x|_{z=b}$ on $z = b$.

$$\underline{U = v_y}: \quad \Psi = n\left(\frac{\rho g}{\eta}\right)^n (h-z)^{n-1} \left[\sqrt{\left(\frac{\partial h}{\partial x}\right)^2 + \left(\frac{\partial h}{\partial y}\right)^2}\right]^{n-1} \frac{\partial h}{\partial y}\left(1 + \frac{\partial \eta}{\partial z}\frac{h-z}{\eta}\right),$$ (A4)





with $\Gamma = 0$ on $z = h$, and $\bar{U} = v_y|_{z=b}$ on $z = b$.

$$\underline{U = v_z}: \quad \Psi = -\frac{\partial^2 v_x}{\partial x \, \partial z} - \frac{\partial^2 v_y}{\partial y \, \partial z}, \tag{A5}$$

with $\Gamma = -(\partial v_x / \partial x)_{z=h} - (\partial v_y / \partial y)_{z=h}$ on $z = h$, and $\bar{U} = v_z|_{z=b}$ on $z = b$.

$$\underline{U = p}: \quad \Psi = 0, \tag{A6}$$

5  with $\Gamma = -\rho g$ on $z = b$, and $\bar{U} = 0$ on $z = h$.

*Acknowledgements.* We wish to thank A. Abe-Ouchi (Univ. Tokyo) and F. Saito (JAMSTEC Yokohama) for helpful discussions, and O. Gagliardini (Univ. Grenoble Alpes) for his contributions to the SIA solver in Elmer/Ice.

HS, RG and SS were supported by a Grant-in-Aid for Scientific Research A (No. 22244058) from the Japan Society for the Promotion of Science (JSPS). HS and RG were supported by a JSPS Grant-in-Aid for Scientific Research A (No. 25241005). HS was further supported by

10  a JSPS Postdoctoral Fellowship for Overseas Researchers (Pathway to University Positions in Japan) (No. PU15902) and an associated JSPS Grant-in-Aid for Postdoctoral Research Fellows (No. 15F15902).



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





**Table 1.** Standard physical parameters used for the simulations with both the FS and SIA dynamics (following SeaRISE Antarctica with SICOPOLIS (Sato and Greve, 2012; Bindschadler et al., 2013)).

| Quantity | Value |
|---|---|
| Density of ice, $\rho$ | $910 \, \mathrm{kg \, m^{-3}}$ |
| Gravitational acceleration, $g$ | $9.81 \, \mathrm{m \, s^{-2}}$ |
| Length of year, $1 \, \mathrm{a}$ | $31\,556\,926 \, \mathrm{s}$ |
| Power law exponent, $n$ | 3 |
| Flow enhancement factor, $E$ | 5 |
| Rate factor, $A(T')$ | $A_0 \, \mathrm{e}^{-Q/R(T_0+T')}$ |
| Pre-exponential constant, $A_0$ | $3.985 \times 10^{-13} \, \mathrm{s^{-1} \, Pa^{-3}} \quad (T' \leq -10°\mathrm{C})$ |
| | $1.916 \times 10^{3} \, \mathrm{s^{-1} \, Pa^{-3}} \quad (T' \geq -10°\mathrm{C})$ |
| Activation energy, $Q$ | $60 \, \mathrm{kJ \, mol^{-1}} \quad (T' \leq -10°\mathrm{C})$ |
| | $139 \, \mathrm{kJ \, mol^{-1}} \quad (T' \geq -10°\mathrm{C})$ |
| Melting temperature at low pressure, $T_0$ | $273.16 \, \mathrm{K}$ |
| Clausius-Clapeyron constant, $\beta$ | $9.8 \times 10^{-8} \, \mathrm{K \, Pa^{-1}}$ |
| Universal gas constant, $R$ | $8.314 \, \mathrm{J \, mol^{-1} K^{-1}}$ |
| Heat conductivity of ice, $\kappa$ | $9.828 \, \mathrm{e}^{-0.0057 \, T[\mathrm{K}]} \, \mathrm{W \, m^{-1} K^{-1}}$ |
| Specific heat of ice, $c$ | $(146.3 + 7.253 \, T[\mathrm{K}]) \, \mathrm{J \, kg^{-1} K^{-1}}$ |
| Latent heat of ice, $L$ | $3.35 \times 10^{5} \, \mathrm{J \, kg^{-1}}$ |





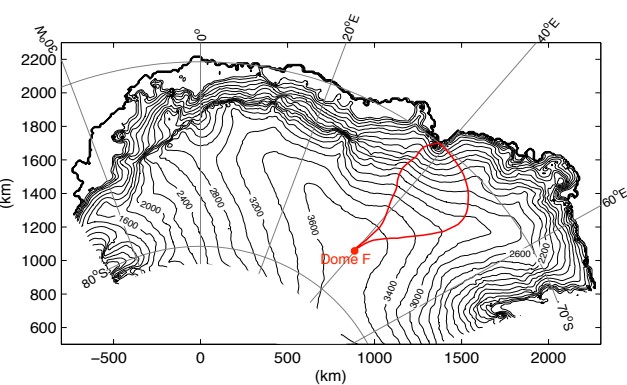

**Figure 1.** Shirase Drainage Basin, delimited by the two red lines. They originate at Dome Fuji and follow the steepest descent of the free surface, hence approximately representing flow lines. Underlying map generated with data by Bamber et al. (2009).



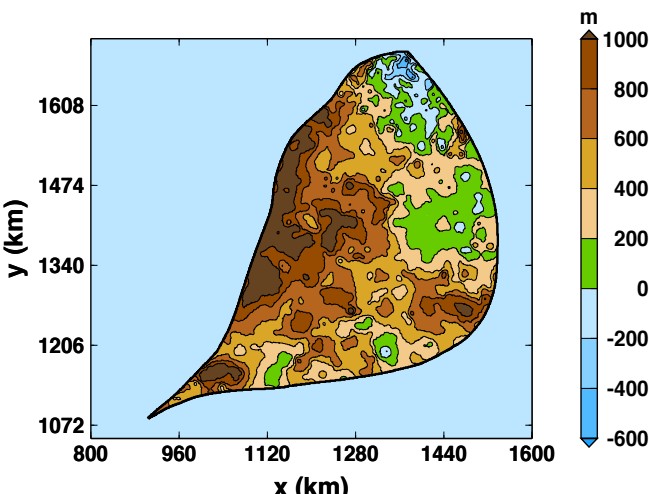

**Figure 2.** Bed topography of the Shirase Drainage Basin (Fretwell et al., 2013).



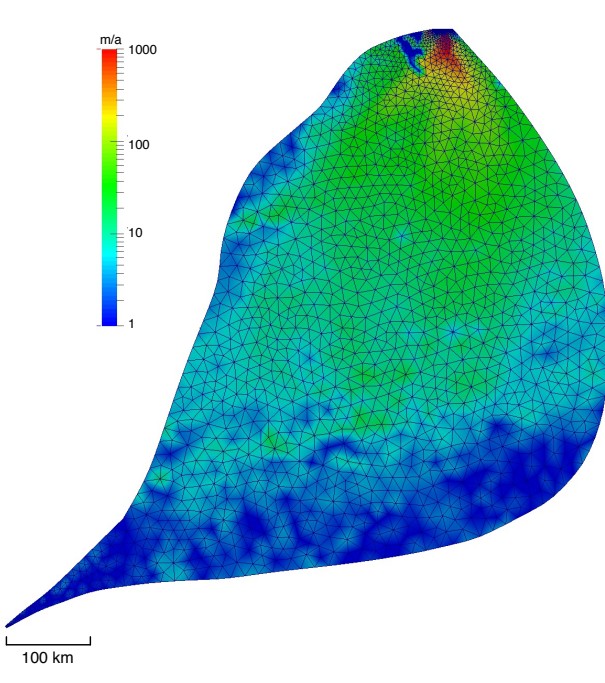

**Figure 3.** Surface velocities by Rignot et al. (2011), finite element mesh of the computational domain superimposed. Note the mesh refinement towards the grounding line.





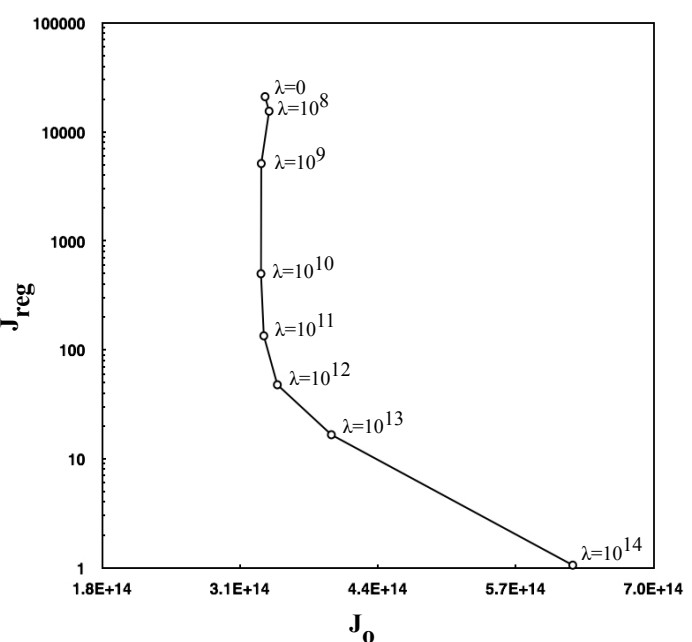

**Figure 4.** L-curve obtained with the control inverse method.





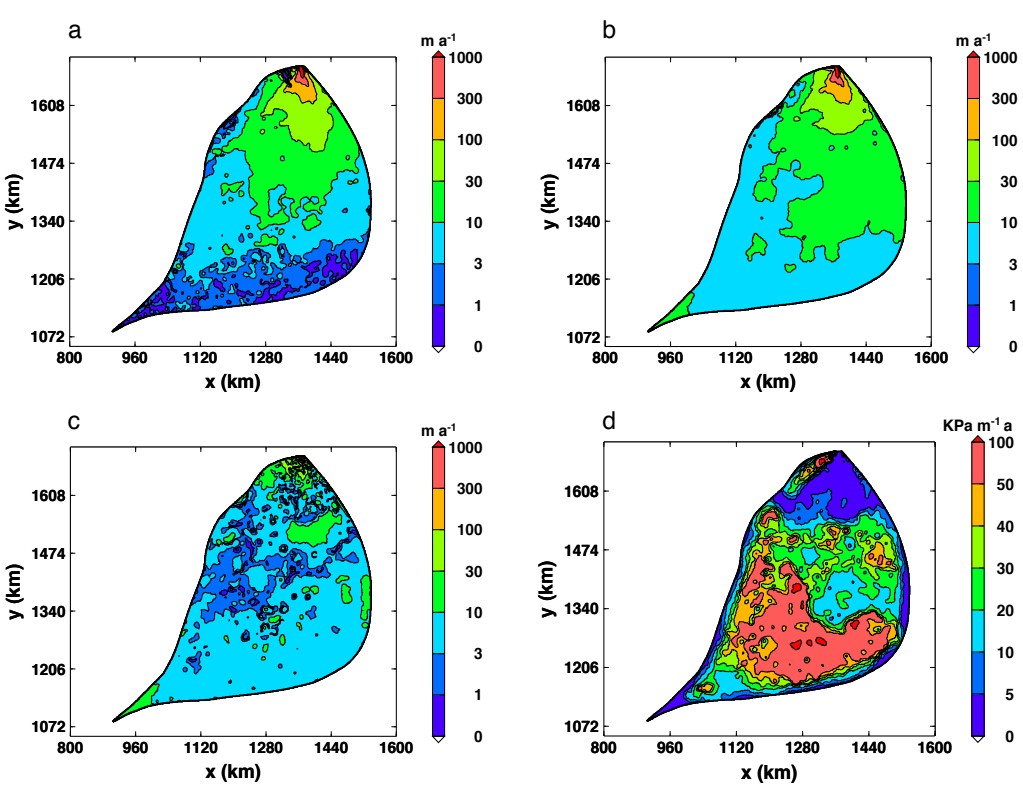

**Figure 5.** (a) Observed, InSAR-based ice surface velocities (Rignot et al., 2011), (b) surface velocities computed using FS and the control inverse method, (c) absolute error of the surface velocities $|\boldsymbol{v}_\mathrm{h} - \boldsymbol{v}_\mathrm{h}^{\mathrm{obs}}|$ at the end of the inversion procedure, and (d) computed basal friction coefficient $\beta$.





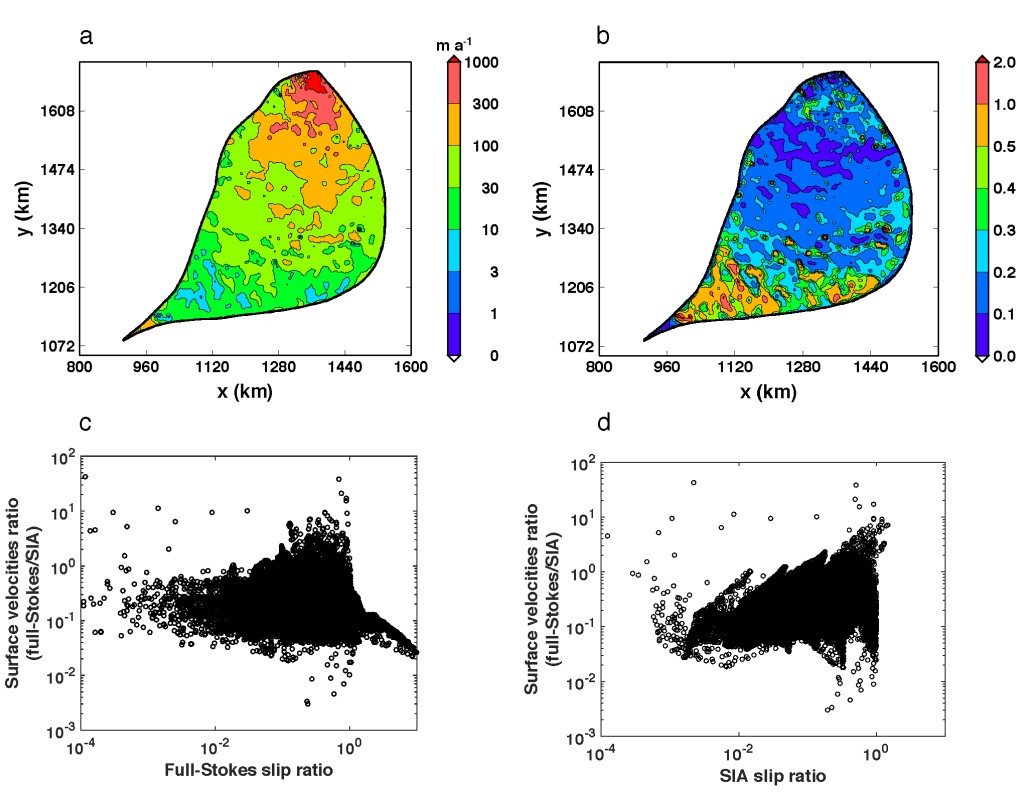

**Figure 6.** (a) Surface velocities computed using SIA applying the basal friction coefficient $\beta$ obtained with the control inverse method (Fig. 5d), (b) surface velocity ratio (FS/SIA), (c) scatter plot of the surface velocity ratio vs. the FS slip ratio, and (d) scatter plot of the surface velocity ratio vs. the SIA slip ratio.





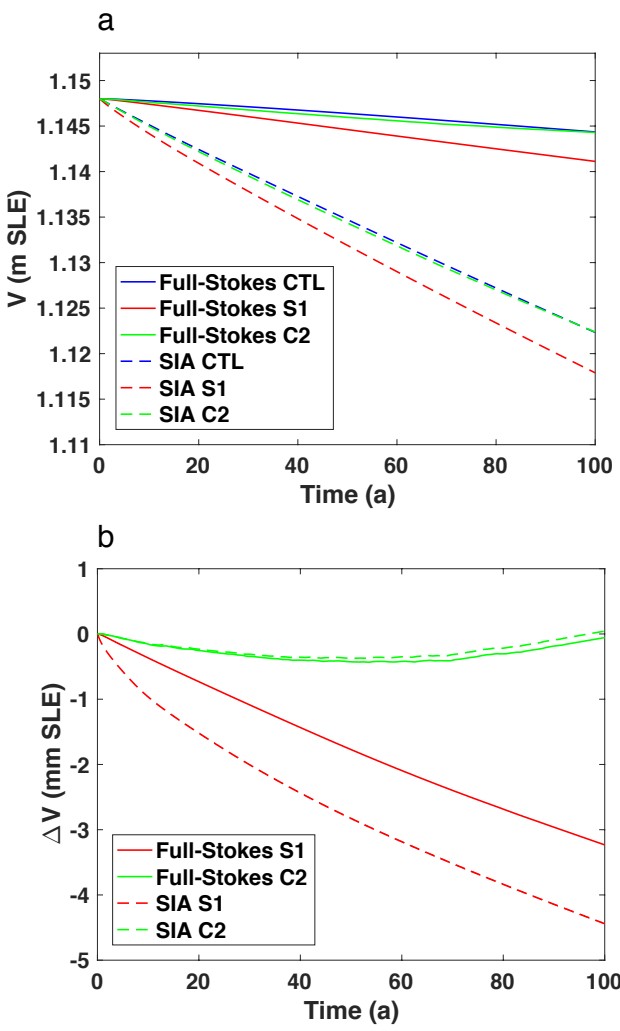

**Figure 7.** (a) Ice volume ($V$) simulated with FS and SIA for experiments CTL (constant-climate control run), S1 (basal sliding experiment) and C2 (surface climate experiment), and (b) ice volume relative to CTL ($\Delta V$) for experiments S1 and C2. The volumes are expressed in sea level equivalents (SLE). $t = 0$ corresponds to the year 2004.





**Figure 8.** Surface velocities (a, c, e) and slip ratios (b, d, f) computed with the FS dynamics for experiment CTL (constant-climate control run; a, b), S1 (basal sliding experiment; c, d) and C2 (surface climate experiment; e, f) at $t = 100$ years (year 2104).





**Figure 9.** Surface velocities (a, c, e) and slip ratios (b, d, f) computed with the SIA dynamics for experiment CTL (constant-climate control run; a, b), S1 (basal sliding experiment; c, d) and C2 (surface climate experiment; e, f) at $t = 100$ years (year 2104).





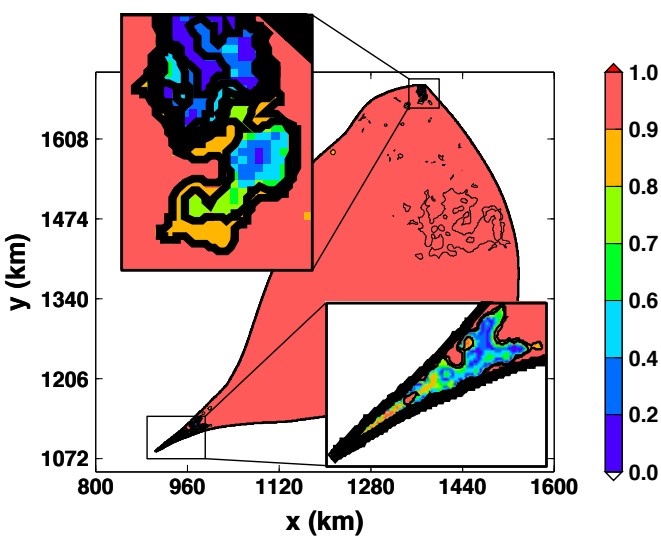

**Figure 10.** Stress ratio $\varsigma = \sqrt{\tau_{xz}^2 + \tau_{yz}^2}/\sigma_{\mathrm{e}}$ computed for the constant-climate control run CTL with FS.