# Peer review of "Regional modeling of the Shirase Drainage Basin, East Antarctica: Full-Stokes vs. shallow-ice dynamics"

_The Cryosphere, 2016_

## Referee Comment (RC1) · Anonymous Referee #1 · 20 Dec 2016

The manuscript "Regional modeling of the Shirase Drainage Basin, East Antarctica: Full-Stokes vs. shallow-ice dynamics" by H. Seddik et al. presents simulations of the Shirase Drainage Basin performed with two different stress balance analyses in order to investigate the impact of the choice of stress balance approximation. The simulations are initialized using an inverse method and full-Stokes stress balance in order to match the observed surface velocities and infer the unknown basal friction. The same basal friction is used for both full-Stokes and the shallow-ice approximation, and simulations based on three different scenarios are run for 100 years.

I am concerned about the novelty of this paper. As stated by the authors in the paper, the Shirase glacier is "one of the fastest flowing glaciers in Antarctica" and its

flow is "dominated by sliding". However, the authors also explain that the shallow-ice approximation "assumes that grounded ice flow is governed only by ice pressure and the vertical shear". Using the shallow-ice approximation for modeling such a glacier is not valid here is therefore absolutely no reason to compare full-Stokes and shallow-ice simulations for this glacier. The shallow-ice approximation has been developed 30 years ago and has been extensively used, but is known to be valid only on slow moving areas where the motion is dominated by vertical shear. Fast flowing glaciers are dominated by basal sliding and lateral shear cannot be neglected as it provides significant resistance to the flow. It was therefore expected that the shallow-ice approximation would not perform well compared to full-Stokes on this glacier. The conclusion of this paper suggesting that "careful consideration must be given to the representation of ice flow physics when attempting to model the dynamics and evolution of ice sheet areas containing ice streams and outlet glaciers" is not novel.

Comparing different ice flow approximations is not new, and has been studied for at least a decade, on a number of idealized geometries (Hindmars„ 2004; Gudmundsson 2008) and real glaciers (Morlighem et al., 2010; Seddik et al., 2012; Furst et al., 2013), so the domain of validity of the different stress balance approximations is well known and there is nothing new added in this paper. Finally, the simulations performed in this manuscript rely on the Elmer/Ice software, that was recently used to develop a dynamical coupling between full-Stokes and the shallow-ice approximation (Ahlkrona et al., 2016). Applying this new coupling method to the Shirase Glacier and comparing its performance and accuracy to a more traditional full-Stokes model would have been of greater interest for this study.

**1 References**

Ahlkrona, J., P. Lotstedt, N. Kirchner, and T. Zwinger, Dynamically coupling the nonlinear Stokes equations with the shallow ice approximation in glaciology: Description and first applications of the ISCAL method, J. Comput. Phys., 308, doi:10.1016/j.jcp.2015.12.025, 2016.

Furst, J. J., H. Goelzer, and P. Huybrechts, Effect of higher-order stress gradients on the centennial mass evolution of the Greenland ice sheet, Cryosphere, 7 (1), 183–199, doi:10.5194/tc-7-183-2013, 2013.

Gudmundsson, G., Analytical solutions for the surface response to small amplitude perturbations in boundary data in the shallow-ice-stream approximation, Cryosphere, 2, 77–93, 2008.

Hindmarsh, R., A numerical comparison of approximations to the Stokes equations used in ice sheet and glacier modeling, J. Geophys. Res., 109, doi:10.1029/2003JF000065, 2004.

Morlighem, M., E. Rignot, H. Seroussi, E. Larour, H. Ben Dhia, and D. Aubry, Spatial patterns of basal drag inferred using control methods from a full-Stokes and simpler models for Pine Island Glacier, West Antarctica, Geophys. Res. Lett., 37, doi:10.1029/2010GL043853, 2010.

Seddik, H., R. Greve, Z. Zwinger, and O. Gillet-chaulet, F.and Gagliardini, Simulations of the Greenland ice sheet 100 years into the future with the full Stokes model Elmer/Ice, J. Glaciol., 58 (209), doi:10.3189/2012JoG11J177, 2012.

---

## Short Comment (SC1) · 23 Dec 2016

**tc-2016-243 – Reply to RC1**

The manuscript "Regional modeling of the Shirase Drainage Basin, East Antarctica: Full-Stokes vs. shallow-ice dynamics" by H. Seddik et al. presents simulations of the Shirase Drainage Basin …

We wish to thank the referee for his/her efforts, even though we largely disagree with the assessment. For details please see below.

I am concerned about the novelty of this paper. As stated by the authors in the paper, the Shirase glacier is "one of the fastest flowing glaciers in Antarctica" and its flow is "dominated by sliding". However, the authors also explain that the shallow-ice approximation "assumes that grounded ice flow is governed only by ice pressure and the vertical shear". Using the shallow-ice approximation for modeling such a glacier is not valid here is therefore absolutely no reason to compare full-Stokes and shallow-ice simulations for this glacier.

Note that we have not only modelled the Shirase Glacier, but the entire drainage basin all the way up to Dome Fuji (see our Fig. 1). This is a huge area of ~ 200,000 km$^2$, large enough to qualify as a stand-alone ice sheet if it were not connected to an even bigger ice body. In most of it, "normal", slow ice flow prevails. The regions where it is a priori clear that the shallow-ice approximation (SIA) is problematic constitute only a very small part of the drainage basin (see, e.g., our Fig. 10). Therefore, we strongly disagree with the referee's statement that using the SIA for modeling this region is not valid from the outset.

The shallow-ice approximation has been developed 30 years ago and has been extensively used …

Not everything that is old is bad! For example, Newtonian mechanics has been around for 300 years, and, even though we learned about its limitations over time, it is still a very useful tool and being used extensively. The negative ring this half-sentence conveys is therefore not appropriate.

… but is known to be valid only on slow moving areas where the motion is dominated by vertical shear. Fast flowing glaciers are dominated by basal sliding and lateral shear cannot be neglected as it provides significant resistance to the flow. It was therefore expected that the shallow-ice approximation would not perform well compared to full-Stokes on this glacier.

No, this is not correct. We would agree if we only modelled the fast-flowing Shirase Glacier itself, but, as stated above, this is not the case. Our modelling study concerns a significant part of the Antarctic ice sheet, in which the majority of the ice flows rather slowly and exhibits a slip ratio of < 0.5 (that is, the amount of basal sliding is limited; confirmed by our Figs. 8 and 9). Under such conditions, we think that it is an interesting and valid test to check how the SIA performs compared to full Stokes (FS).

The great strength of SIA is its simplicity and enormous computational efficiency. Even in connection with the shelfy stream approximation as hybrid models, SIA is often the only viable alternative in terms of computing resources for applications covering large spatial and temporal scales, such as paleoclimatic runs of an entire ice sheet. Showing by a case study, like ours for the Shirase drainage basin, where exactly the deviations of SIA results from FS exceed an acceptable margin is something that adds valuable information to potentially existing simulation challenges.

The conclusion of this paper suggesting that "careful consideration must be given to the representation of ice flow physics when attempting to model the dynamics and evolution of ice sheet areas containing ice streams and outlet glaciers" is not novel.

We are fully aware that this particular conclusion is not new, but rather a confirmation of previous findings. That's why we started the sentence with "This confirms that…" (page 1, line 10). In order to make it even clearer, we can change it to something like "This confirms findings of earlier studies that…", or delete the sentence altogether.

Comparing different ice flow approximations is not new, and has been studied for at least a decade, on a number of idealized geometries (Hindmars,, 2004; Gudmundsson 2008) and real glaciers (Morlighem et al., 2010; Seddik et al., 2012; Furst et al., 2013), so the domain of validity of the different stress balance approximations is well known and there is nothing new added in this paper.

We did not wish to imply that our paper is the first ever in which different force balances have been compared, and we apologize for not having mentioned more of these earlier studies in the introduction (so far, we only cite Morlighem et al. (2010)). This can be fixed in a revised version. However, the statement that "there is nothing new added in this paper" is definitely not true. As the referee points out, Hindmarsh (2004) and Gudmundsson (2008) dealt only with idealized geometries. Seddik et al. (2012) compared FS and SIA for SeaRISE-Greenland scenarios, but used two different models, namely Elmer/Ice for FS and SICOPOLIS for SIA, which limits the comparability (because differences can also arise from different numerics etc.). Fürst et al. (2013), also a study on the entire Greenland ice sheet, used only one model, but the five approximations to the force balance are between (and including) SIA and Blatter–Pattyn (aka first-order approximation), thus not including FS.

The study by Morlighem et al. (2010) is probably the most similar one to ours because it dealt with a part of Antarctica, used different force balances within one model and a control method to infer basal drag. However, there are still major differences. It dealt with a much smaller area, namely the Pine Island Glacier and its immediate vicinity. Therefore, the characteristics of their domain is very different from ours in that it contains a much larger fraction of fast-flowing ice. For such a domain, the SIA would be clearly inappropriate. Consequently the authors didn't use it and compared FS, Blatter–Pattyn and the shelfy stream approximation. Further, the study only investigated present-day stress and velocity fields, while we also discuss time-dependent future climate scenarios.

In short, our study is (to our best knowledge) the first in which FS and SIA are compared within one model for an application to a large area that has the characteristics of an entire ice sheet. We think that this is sufficiently novel to make our paper a valuable contribution.

Finally, the simulations performed in this manuscript rely on the Elmer/Ice software, that was recently used to develop a dynamical coupling between full-Stokes and the shallow-ice approximation (Ahlkrona et al., 2016). Applying this new coupling method to the Shirase Glacier and comparing its performance and accuracy to a more traditional full-Stokes model would have been of greater interest for this study.

Frankly, we think that such a suggestion exceeds the role of a referee (which is to evaluate a given paper, rather than suggesting something more or less completely different). Anyway, we hope that our above arguments are sufficient to explain why we did what we did, and why we think that our study is interesting for the community.

---

## Referee Comment (RC2) · Anonymous Referee #2 · 4 Jan 2017

The manuscript provides a valuable examination of the ice sheet volume change for an East-Antarctic basin under selected scenarios from the SeaRISE effort, using two formulation for ice dynamics: Full-Stokes (FS) versus Shallow Ice (SIA).

General comments:

The strength of the study is that it attempts to maximize the similarities between the numerical simulations: the mesh is the same, same distribution of basal friction coefficient, same model (Elmer/Ice) etc in order to allow a clean comparison between the FS and SIA solutions. The major finding is a confirmation that the choice of ice dynamic will impact the ice volume evolution. Although this is not ground breaking, the value in the work is that it is a step towards understanding the sources of uncertainty in ice

sheet evolution and hence sea level projections.

The manuscript is very well written and has a clear structure. The discussion and conclusion addressed many of the questions that came to my mind when I was reading the results. The tables and figures used are necessary and well prepared (apart from what is noted in the minor comments).

My major criticism is that the study would have been more interesting/complete/valuable if additional solver that seems to be available within the Elmer/Ice toolkit had been used too. In particular since it is recognized in the community that SIA is not ideal for capturing Antarctic ice sheet dynamics. Nonetheless the authors do acknowledge this limitation and the point is raised in the discussion as further work.

Minor comments:

P8, L10: Could you add an explanation of the need to set a minimum thickness of 10 m?

Figure 1: You have space in the figure to write "Fuji" in full.

Figure 4 caption: can you improve the caption for the readers that like to look at figures & caption without having to dig in the text for understanding? Ie: the main text explains what the axis are but the caption by itself is not very meaningful.

Figure 6: either in the figure or in the caption: can you define what the slip ratio is? It is defined in the introduction, but a reader may have forgotten about this.

---

## Author Response (AR1)

**tc-2016-243 – Reply to Reviewers**

Please find below a detailed, point-to-point answer to both RC1 and RC2, in which the reviewers' comments appear in red, while our replies are written in black. We are attaching a diff file that shows in detail the changes we made in the manuscript.

**Reply to RC1**

The manuscript "Regional modeling of the Shirase Drainage Basin, East Antarctica: Full-Stokes vs. shallow-ice dynamics" by H. Seddik et al. presents simulations of the Shirase Drainage Basin …

We wish to thank the reviewer for his/her efforts, even though we largely disagree with the assessment. For details please see below.

I am concerned about the novelty of this paper. As stated by the authors in the paper, the Shirase glacier is "one of the fastest flowing glaciers in Antarctica" and its flow is "dominated by sliding". However, the authors also explain that the shallow-ice approximation "assumes that grounded ice flow is governed only by ice pressure and the vertical shear". Using the shallow-ice approximation for modeling such a glacier is not valid here is therefore absolutely no reason to compare full-Stokes and shallow-ice simulations for this glacier.

Note that we have not only modelled the Shirase Glacier, but the entire drainage basin all the way up to Dome Fuji (see our Fig. 1). This is a huge area of $\sim$ 200,000 km$^2$, large enough to qualify as a stand-alone ice sheet if it were not connected to an even bigger ice body. In most of it, slow, vertical-shear-dominated ice flow prevails (now mentioned in the beginning of the introduction). The regions where it is a priori clear that the shallow-ice approximation (SIA) is problematic constitute only a very small part of the drainage basin (see, e.g., our Fig. 10). Therefore, we strongly disagree with the reviewer's statement that using the SIA for modeling this region is not valid from the outset.

The shallow-ice approximation has been developed 30 years ago and has been extensively used …

Not everything that is old is bad! For example, Newtonian mechanics has been around for 300 years, and, even though we learned about its limitations over time, it is still a very useful tool and being used extensively. The negative ring this half-sentence conveys is therefore not appropriate.

… but is known to be valid only on slow moving areas where the motion is dominated by vertical shear. Fast flowing glaciers are dominated by basal sliding and lateral shear cannot be neglected as it provides significant resistance to the flow. It was therefore expected that the shallow-ice approximation would not perform well compared to full-Stokes on this glacier.

No, this is not correct. We would agree if we only modelled the fast-flowing Shirase Glacier itself, but, as stated above, this is not the case. Our modelling study concerns a significant part of the Antarctic ice sheet, in which the majority of the ice flows rather slowly and exhibits a slip ratio of < 0.5 (that is, the amount of basal sliding is limited; confirmed by our Figs. 8 and

9). Under such conditions, we think that it is an interesting and valid test to check how the SIA performs compared to full Stokes (FS).

The great strength of SIA is its simplicity and enormous computational efficiency. Even in connection with the shelfy stream approximation as hybrid models, SIA is often the only viable alternative in terms of computing resources for applications covering large spatial and temporal scales, such as paleoclimatic runs of an entire ice sheet. Showing by a case study, like ours for the Shirase drainage basin, where exactly the deviations of SIA results from FS exceed an acceptable margin is something that adds valuable information to potentially existing simulation challenges.

The conclusion of this paper suggesting that "careful consideration must be given to the representation of ice flow physics when attempting to model the dynamics and evolution of ice sheet areas containing ice streams and outlet glaciers" is not novel.

We are fully aware that this particular conclusion is not new, but rather a confirmation of previous findings. That's why we started the sentence with "This confirms that…". In order to make it even clearer, we have changed it to "This confirms findings of earlier studies that…" (page 1, line 10), and we have deleted the last paragraph of the conclusion.

Comparing different ice flow approximations is not new, and has been studied for at least a decade, on a number of idealized geometries (Hindmars,, 2004; Gudmundsson 2008) and real glaciers (Morlighem et al., 2010; Seddik et al., 2012; Furst et al., 2013), so the domain of validity of the different stress balance approximations is well known and there is nothing new added in this paper.

We did not wish to imply that our paper is the first ever in which different force balances have been compared, and we apologize for not having mentioned more of these earlier studies in the introduction. However, the statement that "there is nothing new added in this paper" is definitively not true. As the reviewer points out, Hindmarsh (2004) and Gudmundsson (2008) dealt only with idealized geometries. Seddik et al. (2012) compared FS and SIA for SeaRISE-Greenland scenarios, but used two different models, namely Elmer/Ice for FS and SICOPOLIS for SIA, which limits the comparability (because differences can also arise from different numerics etc.). Fürst et al. (2013), also a study on the entire Greenland ice sheet, used only one model, but the five approximations to the force balance are between (and including) SIA and Blatter–Pattyn (aka first-order approximation), thus excluding FS.

The study by Morlighem et al. (2010) is probably the most similar one to ours because it dealt with a part of Antarctica, used different force balances within one model and a control method to infer basal drag. However, there are still major differences. It dealt with a much smaller area, namely the Pine Island Glacier and its immediate vicinity. Therefore, the characteristics of their domain is very different from ours in that it contains a much larger fraction of fast-flowing ice. For such a domain, the SIA would be clearly inappropriate. Consequently the authors didn't use it and compared FS, Blatter–Pattyn and the shelfy stream approximation. Further, the study only investigated present-day stress and velocity fields, while we also discuss time-dependent future climate scenarios.

In short, our study is (to our best knowledge) the first in which FS and SIA are compared within one model for an application to a large area that has the characteristics of an entire ice sheet (slow flow in the interior [= largest part of the area], fast flow near the grounding line). We

think that this is sufficiently novel to make our paper a valuable contribution, and we have revised the introduction significantly to make this point clear.

In addition, we would like to point out that, beyond the FS-to-SIA comparison, modelling the Shirase Drainage Basin in 3D with FS is novel in itself, and it is relevant because the area is a focus of Japanese research activities in Antarctica. Some findings of the FS simulations, all reported in the paper, are: (1) the observed surface velocity distribution can be reproduced well (Sect. 4.1), (2) the basal friction generally decreases towards the grounding line (in line with expectations; Sect. 4.1), (3) the simulated and observed present-day net mass balances agree within the observational uncertainty (Sect. 4.2), (4) the sensitivity of the Shirase Drainage Basin to SeaRISE-type basal sliding and surface climate experiments is similar to the sensitivity of the entire Antarctic ice sheet (Sect. 5). These findings are interesting in itself and will also attract the attention of colleagues who are just interested in the dynamics of the region, even if they do not care so much about the FS-vs.-SIA story.

Finally, the simulations performed in this manuscript rely on the Elmer/Ice software, that was recently used to develop a dynamical coupling between full-Stokes and the shallow-ice approximation (Ahlkrona et al., 2016). Applying this new coupling method to the Shirase Glacier and comparing its performance and accuracy to a more traditional full-Stokes model would have been of greater interest for this study.

Frankly, we think that this suggestion goes a bit too far because it would be something more or less completely different. Anyway, we hope that our above arguments are sufficient to explain why we did what we did, and why we think that our study is interesting for the community.

**Reply to RC2**

The manuscript provides a valuable examination of the ice sheet volume change for an East-Antarctic basin under selected scenarios from the SeaRISE effort, using two formulation for ice dynamics: Full-Stokes (FS) versus Shallow Ice (SIA).

General comments:

The strength of the study is that it attempts to maximize the similarities between the numerical simulations: the mesh is the same, same distribution of basal friction coefficient, same model (Elmer/Ice) etc in order to allow a clean comparison between the FS and SIA solutions. The major finding is a confirmation that the choice of ice dynamic will impact the ice volume evolution. Although this is not ground breaking, the value in the work is that it is a step towards understanding the sources of uncertainty in ice sheet evolution and hence sea level projections.

The manuscript is very well written and has a clear structure. The discussion and conclusion addressed many of the questions that came to my mind when I was reading the results. The tables and figures used are necessary and well prepared (apart from what is noted in the minor comments).

We wish to thank the reviewer for his/her efforts, and for the positive assessment of our work.

My major criticism is that the study would have been more interesting/complete/valuable if additional solver that seems to be available within the Elmer/Ice toolkit had been used too. In particular since it is recognized in the community that SIA is not ideal for capturing Antarctic

ice sheet dynamics. Nonetheless the authors do acknowledge this limitation and the point is raised in the discussion as further work.

Along this line, we would prefer to leave it with the current FS-vs.-SIA comparison. Of course, any scientific study has room for adding more experiments/analysis etc. However, we feel that what we have now should suffice to be of interest for the community.

Minor comments:

P8, L10: Could you add an explanation of the need to set a minimum thickness of 10 m?

A minimum thickness is required in order to avoid having 2D (rather than 3D) elements. 2D elements would be treated as degenerated elements during the finite element assembly, so that the assembly would fail. The reason why we chose 10 m is to avoid a too low aspect ratio (thickness-to-width ratio) of the finite elements, which can cause the numerical solution to become unstable. We have added this explanation at the end of Section 2.4.

Figure 1: You have space in the figure to write "Fuji" in full.

Done (now Fig. 1a).

Figure 4 caption: can you improve the caption for the readers that like to look at figures & caption without having to dig in the text for understanding? Ie: the main text explains what the axis are but the caption by itself is not very meaningful.

We have extended the caption to make it more self-explanatory (now Fig. 2).

Figure 6: either in the figure or in the caption: can you define what the slip ratio is? It is defined in the introduction, but a reader may have forgotten about this.

We have added the definition in the caption (now Fig. 4).

**Further changes**

In response to an earlier comment from the scientific editor, we have reduced the number of figures by combining the previous Figs. 1–3 into the new Fig. 1.

We have changed the symbol used for the Clausius–Clapeyron constant in Table 1 from $\beta$ to $\Delta T_\mathrm{m}/\Delta p$ in order to avoid the double use of $\beta$ (denotes also the basal friction coefficient).

---

## Referee Report (RR1)

April 2, 2017

The manuscript "Regional modeling of the Shirase Drainage Basin, East Antarctica: Full-Stokes vs. shallow-ice dynamics" presents model results on Shirase Drainage Basin produced by different methods of force balance approximations (full-Stokes and Shallow-Ice Approximation). The basal friction coefficient for both force methods is obtained with FS dynamics based on the control inverse method and the observed velocity data. Experiments 100 years into the future are taken with different scenarios defined by the SeaRISE project. The conclusion is that FS is superior to SIA in the area.

General comments:

The manuscript is easy to follow and well organized.

There are quite a few of model inter-comparison studies on ideal and real ice sheets and have already showed the insufficiency of SIA, hence the conclusion is not novel. However, comparing to the former works, this study maximally exclude influence of factors other than force balance approximation by using the same settings in Elmer/Ice. On this typical drainage basin, this study quantified the importance of longitude and lateral stress in different regions. It can be taken as a reference for ice dynamical studies of such regions in East Antarctica.

Minor comments:

P2L20 shows that Seddik et al., (2012) found Elmer/Ice with FS was more sensitive than SICOPO-LIS with SIA for the double basal sliding experiment, which is not the case in your study. Can you discuss the possible explanations of the difference?

P10L21-22: The sentence is confusing to me. In general, SIA produce higher surface velocity in your simulations, so surface velocity ratio (FS/SIA) < 1 in general. When the ratio is less than 1, increasing of the ratio means SIA produces velocities nearer to FS, which means the higher performance of SIA. Then this sentence means SIA performs better in slipperier region, which is contradict with the following sentence.

Definition of T' and T in eqs 4 and 5 are missed.

---

## Referee Report (RR2)

**Review of "Regional modeling of the Shirase Drainage Basin, East Antarctica: Full-Stokes vs. shallow-ice dynamics" by Seddik and colleagues**

April 25, 2017

This study made by Seddik and colleagues investigates the behaviour of the Shirase drainage basin in East Antarctica, using two sets of equations for the stress balance, thus comparing the most sophisticated full-Stokes (FS) model to the Shallow Ice Approximation (SIA). Prior to transient simulations, both models are initialized by applying an inverse method to the full-Stokes equations, which results in a unique basal friction field used to initialize both the FS and SIA models. The authors then perform a series of transient simulations using some of the SeaRISE experiments. The two models are suposedly based on the same numerics as both included of the finite element software Elmer/ice.

I was asked to review this paper with the specific focus on its novelty. After reading carefully both the papers and the discussion, the only novelty that I can see is the use of the same software (or numerics) and grid for the two representations of the stress balance equation. To my knowledge, the effect of different numerics for this kind of setup has been poorly investigated in the past. I have however the study in mind from Pattyn and Durand (2013) who investigated the results of the MISMIP3D experiments with a focus on the approximation used. They show for instance that all the SSA models behave similarly in the frame of those ideal experiments. For the rest, the fact that SIA gives faster velocities because of the lack of longitudinal and lateral shearing stress, was already shown by, e.g., Le Meur et al. (2004) and, as the authors say, Seddik et al. (2012).

This being said, I have a major criticism of the paper. If the purpose was to reduce the effect of the numerics, I think the way you do your experiments may strongly bias the results, thus vanishing the effect of getting rid of potential numerical issues:

- The initial field of basal friction that the authors use to initialise the SIA model is obtained by applying the inverse method to the FS model. The basal friction field is thus a function of the complete stress tensor, not only the vertical shearing terms (as accounted for in the SIA). While a major assumption is made here, the authors do not discuss the implications or try to show the validity of this approach. For instance, I am quite surprised to see that, while the ratio between the effective strain rate (Figure 10 in the paper) of SIA and FS models is different than the unity only in a limited area of the Shirase basin (the last 20 km or so from the grounding line downstream) the difference between both field of surface velocity is much more extended (up to about 500 upstream of the grounding line, this difference is significant). Looking at the paper from Seddik et al. (2012), the differences between Sicopolis and the FS model of Elmer/Ice seem to be much less spread upstream. In that previous study, the friction law was not relying on the FS equations only, thus it makes me think that the initial field of basal friction is partly responsible for that.

- Also, there is another assumption made that worries me, which is the use of different boundary conditions at the grounding line, either the ocean pressure for the FS model or a constant thickness for the SIA model. This is again something that is not discussed at all, while this crucial boundary is the gate to the ocean and of significant influence on the ice discharge. At least the authors should show an experiment to prove that this assumption has no or little influence on the discharge twoards the ocean.

I have a couple of other critiscims:

- There is no relaxation between the inversion and the predictive simulations. This is made by most numerical studies following an inversion (e.g. Gillet-Chaulet et al., 2012; Cornford et al., 2015) to make sure the initial state is not affected by short-wavelength and large amplitude variation of the surface elevation change rate, which may arise from different resolutions or inconsistencies between the data and the model. If you don't have any, I would be surprised, but it needs to be mentioned and the initial dhdt need to be quantified.

- The domain is delimited with two lateral flow lines, for which your boundary condition leads to high sliding coefficients. I don't think this is an issue for the most upstream part of the drainage basin (say the ice divide), but this is more worrysome in the vicinity of the grounding line, where it can significantly affect your discharge towards the ocean.

**References**

Cornford, S. L., Martin, D. F., Payne, A. J., Ng, E. G., Le Brocq, A. M., Gladstone, R. M., Edwards, T. L., Shannon, S. R., Agosta, C., Van Den Broeke, M. R., Hellmer, H. H., Krinner, G., Ligtenberg, S. R. M., Timmermann, R., and Vaughan, D. G. (2015). Century-scale simulations of the response of the West Antarctic Ice Sheet to a warming climate. *Cryosphere*, 9(4):1579–1600.

Gillet-Chaulet, F., Gagliardini, O., Seddik, H., Nodet, M., Durand, G., Ritz, C., Zwinger, T., Greve, R., and Vaughan, D. G. (2012). Greenland ice sheet contribution to sea-level rise from a new-generation ice-sheet model. *Cryosphere*, 6(6):1561–1576.

Le Meur, E., Gagliardini, O., Zwinger, T., and Ruokolainen, J. (2004). Glacier flow modelling: A comparison of the Shallow Ice Approximation and the full-Stokes solution. *Comptes Rendus Physique*, 5(7):709–722.

Pattyn, F. and Durand, G. (2013). Why marine ice sheet model predictions may diverge in estimating future sea level rise. *Geophysical Research Letters*, 40(16):4316–4320.

Seddik, H., Greve, R., Zwinger, T., Gillet-Chaulet, F., and Gagliardini, O. (2012). Simulations of the Greenland ice sheet 100 years into the future with the full Stokes model Elmer/Ice. *Journal of Glaciology*, 58(209):427–440.

---

## Editor Decision (ED1)

MS 300-321C
4800 Oak Grove Drive
Jet Propulsion Laboratory
Pasadena, CA 91109-8099, U.S.A.
Tel (818) 970-8032
email: eric.larour@jpl.nasa.gov

July 6, 2017

Dear authors:

Thank you for your revised manuscript entitled " **Regional modeling of the Shirase Drainage Basin, East Antarctica: Full-Stokes vs. shallow-ice dynamics**" that you submitted for publication in *The Cryosphere*.

As you may know, I requested a second round of review to address my specific concern regarding originality of the manuscript, as many papers have already been published concerning comparisons between SIA and Higher-Order formulations. This second round of reviews convinced me of the advantage of carrying out such comparison on equivalent framework and grid, which you did here. This concern being addressed, I believe the manuscript is acceptable for publication. However, I would like to see the concerns raised by reviewer #2 of the second round be addressed, in particular as they relate to:

- 1: the inversion of basal friction carried out using the FS formulation and then directly applied to the SIA formulation. Either a new SIA inversion should be carried out, or a serious discussion of the consequences of such an approach should be carried out.
- 2: the nature of icefront boundary conditions being significantly different between both formulations. I would like to see this discussed thoroughly, as it is an intrinsic result of the formulations themselves.
- 3: the absence of relaxation: this should also be thoroughly discussed.

Once these points are addressed, the manuscript will be published. I will carry out an editor review prior to final publication.

Dr. Eric Larour,
Editor *The Cryosphere*

Sincerely yours,

---

## Author Response (AR2)

**tc-2016-243 – Reply to Editor**

As you may know, I requested a second round of review to address my specific concern regarding originality of the manuscript, as many papers have already been published concerning comparisons between SIA and Higher-Order formulations. This second round of reviews convinced me of the advantage of carrying out such comparison on equivalent framework and grid, which you did here. This concern being addressed, I believe the manuscript is acceptable for publication.

Thanks a lot for the positive assessment!

However, I would like to see the concerns raised by reviewer #2 of the second round be addressed, in particular as they relate to:

• 1: the inversion of basal friction carried out using the FS formulation and then directly applied to the SIA formulation. Either a new SIA inversion should be carried out, or a serious discussion of the consequences of such an approach should be carried out.

We extended the previously existing discussion on this matter. It is now in Sect. 5, page 13, lines 16–29.

• 2: the nature of ice front boundary conditions being significantly different between both formulations. I would like to see this discussed thoroughly, as it is an intrinsic result of the formulations themselves.

We already touched upon this issue shortly in Sect. 5. The discussion has now been extended (page 13, line 32 – page 14, line 7).

• 3: the absence of relaxation: this should also be thoroughly discussed.

We added a discussion on this issue in Sect. 5 (page 13, lines 7–15).

Once these points are addressed, the manuscript will be published. I will carry out an editor review prior to final publication.

For your convenience, we are attaching a diff file that shows in detail the changes we made in the manuscript.

[revised manuscript text omitted]